# Applying Universal Principles of ‘Best Interest’: Practice Challenges across Transnational Jurisdictions, Cultural Norms, and Values

**DOI:** 10.3390/children10030537

**Published:** 2023-03-10

**Authors:** Brian Littlechild, Carolyn Housman

**Affiliations:** 1Brian Littlechild, School of Health and Social Work, University of Hertfordshire, Hatfield AL10 9AB, UK; 2Children and Families across Borders, London SW1W 9SA, UK

**Keywords:** kinship care, international placements, child protection, trauma, children’s needs, transnational children’s work, best interests

## Abstract

This article sets out key issues in determining and upholding the best interests of children, in need of social service support, who have family networks that span outside of the UK. These issues are then analysed against whether and how child protection professionals take these into account along with an overall consideration of the United Nations Convention on the Rights of the Child’s (UNCRC) ‘best interests of the child’, when assessing and planning for those needs in kinship care cases. Building on these themes, the findings of an exploratory study on international kinship care cases carried out by Children and Families Across Borders (CFAB), the UK branch of the non-governmental organisation, International Social Service, as well as CFAB’s associated Freedom of Information Requests to the UK government, are examined. These are then analysed in relation to legal and policy documents in England. Agency case records are analysed to identify a range of factors for children placed with ‘kinship’ carers across national borders, relating to the cultural relativity of the ‘best interest’ principle, the availability of family support in different social service structures, the understanding and application of legislation and policy in transnational contexts, and the availability of markers to track and analyse the scale of children crossing borders to join family.

## 1. Introduction

Children are moving across international borders today at greater rates than any seen since the 1939–1945 Second World War [1,2]. In 2020, there were 281 million international migrants; 36 million of these were children [3]. In March 2020, 9.5 million people in the UK had been born abroad [2]. The number of children being born into multi-national families is increasing, and more children are separated from their families by international borders. In 2019, over 25% (3,839,000) of children below the age of 18 residing within the UK had a parent that had been born abroad and were likely to have family there [4]. The recognition of the relevance of migration and the increased attention to comparative welfare in the social policy literature have both had a bearing on the development of ideas regarding international social work in relation to such matters [5,6,7,8].

In pursuit of identifying issues arising from such movements for children, this article sets out key matters for child protection professionals to take into account in situations where a child is taken into state care, but family members overseas may be able to take them. In particular, the analysis looks at four distinct but interlinked issues for such children, acknowledging the intersectionality and compound effects of (1) determining best interests across different cultural norms; (2) evaluating support available for children across different welfare systems; (3) navigating different legal and jurisdictional frameworks; and (4) the viability of data (and tracking) on children who cross international borders. These matters are analysed against whether and how professionals working to ensure those children’s well-being and security take these into account, within an overall consideration of the United Nations Convention on the Rights of the Child’s (UNCRC) ‘best interests of the child’, when assessing and planning for those needs in kinship care cases. Building on these themes, the outcomes of an original study on international kinship care cases carried out by the non-governmental organisation Children and Families Across Borders (CFAB) (UK branch of International Social Service), as well as CFAB’s associated Freedom of Information request to the UK government, are examined. These are then analysed in relation to legal and policy documents in England.

## 2. Literature Review

This section examines the literature in relation to definitions in law and in kinship care policies, the value of kinship care as a form of alternative care, and the literature in relation to the needs and rights of children when there are plans for them to be placed with family overseas. The literature review was based upon both research literature and grey literature, in particular from the UNCRC, and government policy and legislation in England in relation to the ‘best interests of the child’ and kinship care.

### 2.1. Defining Kinship Care

There is no agreed legal definition of ‘kinship care’ in the UK. Indeed, the 2022 Final Report of the Independent Review of Children’s Social Care recommended that ‘there needs to be a much clearer definition of what we mean by kinship care’ [9]. The term ‘kinship care’ is frequently used in the US and the UK, as well as other countries, when children are brought up by extended family members or significant others [10,11,12]. Such placements are often explored when the parents cannot care for the child(ren), and are frequently judged as providing greater security and better outcomes for them [11,13,14].

Kinship care may be a permanent arrangement, usually orchestrated through the use of a legal order or an informal arrangement made privately amongst family [15].

### 2.2. The Case for Kinship Care

Kinship care has been arranged informally throughout recorded history and is now recognised as having many advantages within the formal structures of child protection, most notably the preservation of family, promotion of cultural identity, and reduced separation trauma [16,17,18,19,20]. Support for children remaining with family (kin) is based on the basic principle that it is in their best interest. The best interests of the child concept comes from the United Nations Convention on the Rights of the Child [21,22]:
‘*In all actions concerning children, whether undertaken by public or private social welfare institutions, courts of law, administrative authorities or legislative bodies, the best interests of the child shall be a primary consideration.*’.[22]: article 3

The UNCRC includes provisions around the child’s needs for health; safety; family relationships; well-being; psychological, physical, and emotional development; identity; freedom of expression; privacy; and agency to form their own views and have them heard. Put simply, the best interests of the child are whatever is best for that individual child [23,24,25,26].

The United Kingdom has ratified the UNCRC and implemented it through various pieces of legislation in the devolved nations, but notably through the Children Act of 1989, the Borders, Citizenship, and Immigration Act 2009, and the Children’s Commissioner Offices of each nation [27]. However, it is notable that in England and Northern Ireland, there is no legal requirement for the government to have due regard to the UNCRC, and there has been no progress towards incorporating the UNCRC into their domestic laws [28].

In policy and practice in England and worldwide, ‘kinship care’ is currently one of the favoured options in the discourse surrounding children needing ‘alternative care’ when they are unable to be cared for by their parents [29,30]. England’s Children Act 1989 requires local authorities and their social workers to give preference to placements with kin carers:
‘*In accordance with Section 22C(7) (of the Children Act 1989), in determining which is the most appropriate placement the local authority must ‘give preference to’ a placement with a connected person i.e., a relative, friend or other person connected with the child.*’.[31]:s.3.4

Statutory guidance under that Act in England states that such placements must be considered, and a suitable kin carer should be sought:
‘*It is important that wider family members are identified and involved as early as possible, as they can play a key role in supporting the child and helping the parents to address identified problems. When problems escalate and children cannot live safely with their parents, local authorities should seek to place children with suitable wider family members where it is safe to do so.*’.[32]:22

Moreover, the UK government’s further statutory guidance in its ‘Working Together to Safeguard Children’ maintains reference to kinship care and states that, where the child has links to a foreign country, a social worker may need to work with colleagues abroad [33]. The UN Guidelines for the Alternative Care of Children was created *‘to support efforts to keep children in, or return them to, the care of their family or, failing this, to find another appropriate and permanent solution, including adoption and kafala of Islamic law’* [34]. This clarifies that the removal of a child from their family should be a ‘measure of last resort’. These same principles apply even when the placement must be coordinated across countries. In relation to international placements of 146 children, both adopted and in residential care, Roman et al. [35] concluded that adoption appeared to be more effective in promoting recovery from attachment disorders after early adversity, whilst residential care seemed to cause negative effects. Although this refers to adoption, the key point for our purposes here is that a family setting is positively indicated over a residential oneThese findings, coupled with research now pointing consistently to a number of advantages of kinship care placements over alternatives such as foster care [36,37,38,39,40,41,42], have led to recent policy developments in favour of kinship care where possible [9]. The founding basis of kinship care is illustrated in Table 1 below.

From the review of the relevant literature, policies, and legal issues, the key issues and knowledge in relation to this issue will now be addressed in relation to the challenges and opportunities of international kinship care.

## 3. Determining Best Interests across Different Cultural Norms and Values

For our purposes in this article, it should be noted that there can be additional challenges of determining ‘best interests’ for children when their permanency planning crosses international borders. This may be both practically, as family may be distributed across various countries and time zones, but also emotionally [43]: The wellbeing and feelings of security for the child due to the issues of moving across international borders to potentially largely unknown people, cultures, and languages must be specifically addressed when considering what is in the child’s best interest. Thus, whilst the initial promotion of relationships with family more often leads to better outcomes for children, there is a dearth of evidence that takes into account the added difficulties of accessing families overseas other than the data collected by Children and Families Across Borders, as is explored later in this article.

The Children Act 1989, Section 1, states that the child’s welfare must be of paramount consideration [31]. While welfare is not defined, a number of court cases have considered the application of Article 3 of the UNCRC in relation to child welfare. According to Regina v. Secretary of State for Education and Employment and others (Respondents), ex parte Williamson (Appellant), and per Baroness Hale, ‘the state is entitled to give children the protection they are given by an international instrument to which the UK is a party [the UNCRC]’ [44]. Moreover, the regulations implemented by the Children Act of 1989 state that ‘The objective of planning for permanence is therefore to ensure that children have a secure, stable and loving family to support them through childhood and beyond and to give them a sense of security, continuity, commitment, identity and belonging’ [31]: s 2.3.

The United Nations (UN) Guidelines for Alternative Care of Children confirm:
‘*Planning for care provision and permanency should be based on, notably, the nature and quality of the child’s attachment to his/her family, the family’s capacity to safeguard the child’s well-being and harmonious development, the child’s need or desire to feel part of a family, the desirability of the child remaining within his/her community and country, the child’s cultural, linguistic and religious background, and the child’s relationships with siblings, with a view to avoiding their separation*’.[34]: s.62

However, the development of transnational knowledge to meet children’s needs and best interests poses challenges for policy and practice [5,7,45,46,47]. These include determining what the UNCRC notion of the best interests of a child may be in the context of societal differences related to religion and culture, as these affect beliefs and practices that inform the care of children internationally across borders—see, e.g., Graham [48], and cases within cultures that are themselves not fixed and static —see e.g., [49,50,51].

Childcare approaches and parenting themselves can be seen to be based on different understandings of the structure, roles, and responsibilities of families and children in different countries [48,49,52,53]. Child protection work itself is partly, if not mostly, determined by societal views regarding what constitutes good childcare and child abuse [54,55], thus affecting the culture and norms regarding the definitions and operationalisation of ideas about childhood and the treatment of children [56].

In relation to social workers’ approaches to work with children and families, there is evidence of how the social construction of children affects social workers’ assessments and interventions, with variations in how children are perceived and treated. This has direct relevance for local and international practice [56], which may affect such assessments nationally and globally, and, as it can be argued, particularly in relation to ethnicity and culture. Such factors are part of the influences on whether and how provisions and social work services for children and families have developed in different countries. Table 2 illustrates the elements of the Welfare Checklist, frequently used by social workers to help consider what will be in a child’s best interest.

If the idea of best interest is nearly universally held as paramount for children, but cultural interpretations of it vary, then a deeper understanding of different cultures is key to assessment and planning approaches, as those responsible for assessing and planning children’s international placements need to understand and positively respond to cultural differences. There must be an ability to appreciate the experiences of, and to communicate and work effectively with, people from different cultures [57,58]. It can be argued that in order to expand our theoretical and practical framework for working with different cultures, a comprehensive understanding of the relations between ‘Self’ and the ‘Other’ is necessary [59]. Ben-Ari and Strier [59] proposed that social workers need to recognise how to respond effectively to people of all different cultures, religions, ethnic backgrounds, social classes, and ‘Other’ diversity factors. Social workers need to recognise and value the worth of individuals, families, and communities, and protect and preserve the dignity of all.

Particularly where there are potentially issues of culture disruption, as with children moved from one culture to another, social workers must be able to confidently carry out culturally sensitive work, recognising and working with the strengths of different cultural practices, while at the same time avoiding cultural relativity, which can put children at risk [56,60]. Whilst it is important to develop systems and practices which facilitate transnational social work when this might be appropriate, comparative literature and individual country studies have confirmed that such assumptions can be based on values influenced by ‘western’ or even Anglo-American cultures which underlie definitions, conventions, and international policy papers. Therefore, the development of transnational social work practices that recognises the significant variations in welfare provisions must take place, most specifically children’s social services and the training and supply of social workers, as well as the political, socioeconomic, and cultural conditions in which such provisions—or a lack thereof—have evolved [5,6,45].

The results of cultural complacency are highlighted in the 2002 case of Victoria Climbié, a young girl from the Ivory Coast who suffered months of abuse by her carers, her great-aunt Marie-Therese Kouao and her boyfriend, Carl Manning, and who was eventually murdered by her carers. The social worker assigned to the case stated, in the inquiry into the eight-year-old’s death, that her assumptions about African–Caribbean families influenced her judgement. Social worker Lisa Arthurworrey, of African–Caribbean descent, said she had assumed Victoria’s timidness in the presence of Kouao and Manning stemmed not from fear, but from her African–Caribbean culture. Such a simplified understanding of parenting practices is highlighted by Okpokiri in asserting that it is a failure of the Western social service system not to understand micro-strategies of Nigerian parenting practices and undermines the best interest of the child [61].

## 4. Evaluating Support Systems for Children and Families across Different Cultures and Norms

There are also differences in the wider societal expectations of families, legal frameworks, and the resources available to families in countries globally. For example, some kinship placements take place in countries where it is not mandatory that girls go to school and/or child marriage is accepted; where health provisions and practices differ from those of the UK; and/or where there is only limited acknowledgement and legislation regarding protection of children’s rights and their best interests. In many cases, there needs to be appreciation of the strengths of the family in the context of societal differences, and a balance must be maintained between considering the needs (best interests) of the child and not negatively judging the parenting (or caring) capacity and support systems in another culture. ‘Case by case decisions’ need to be made in relation to children for whom kinship care abroad might be one option that are similar to the decisions which social workers make about ‘domestic placements’. Good practice here involves proper and well-informed preparation, support, and review.

The concept of kinship care is significantly affected by cultural factors resulting in different policies and support practices, for example, in relation to the status of orphans, illegitimacy, abandonment, and adoptions—secret and customary—in some Muslim societies [62]. Additionally, in relation to international kinship care, there are differences in the range of welfare provision—or lack thereof—between the ‘sending countries’ and the ‘receiving countries’. These can be related to economic factors but also to political ideology and national cultures, and such differences are evident within and between continents. For instance, Kriz and Skivenes [63] have explored the challenges that marginalised minority parents face when raising their children in England, Norway, and the United States. The authors found that, while there are similar patterns of challenges across the three countries, including cultural differences, lack of language proficiency, and knowledge of the society and systems, there were also cross-country differences, mainly corresponding to the different aims of the welfare systems and services provided.

## 5. Recognising and Enforcing Carer Responsibilities across Different Judicial Systems

The legal rights and responsibilities attached to the care of a child will vary according to the type of care (adoption, kinship care, legal guardian, foster care, etc.) and this is multiplied when applied to multiple jurisdictions. At a minimum, legal considerations will need to be made in the sending country and the receiving country, but this may also include third countries if the child or family are nationals of another state. For example, although the institution of Kafalah is growing in recognition by many receiving countries, its origin, meaning, and the variety of practices within the Muslim world are unfamiliar to many Western professionals. Within the Muslim world, international definitions are rejected and must be interpreted at a national level. Kafalah is usually defined as (e.g., article 116, Family Code of Algeria) “*the commitment to voluntarily take care of the maintenance, of the education and of the protection of a minor, in the same way as a father would do it for his son*” [64]. However, Iran, Mauritania, and Egypt reject that Kafala is the equivalent of adoption, and in these countries, children without their parents cannot be placed internationally with someone who is not their blood relative. Tunisia and Indonesia allow for full adoption internationally, as long as the adoptive carers are of the same religion as the child [64].

The dangers of teams and courts providing child services not recognising the challenges of different legal and judicial systems are poignantly reviewed in the 2019 case Re: K, T, and U (Placement of Children with Kinship Carers Abroad). Recorder Samuels QC notes:
“*It is well recognised that placing children with kinship carers can represent an optimal outcome for families where it is not safe to permit the children to return to the care of a parent. For example, s.33C(7) of the 1989 Act provides that in determining which is the most appropriate placement, a local authority must ‘give preference’ to a placement with a relative, friend or other connected person. Article 20 of the UN Convention on the Rights of the Child provides that, ‘3… When considering solutions, due regard shall be paid to the desirability of continuity in a child’s upbringing and to the child’s ethnic, religious, cultural and linguistic background.’ However, achieving a lawful, safe and supported placement for children with family or friends abroad represents one of the greatest family law challenges for professionals, lawyers and judges.*”.[65]

Beyond the challenges of different legal frameworks and jurisdictions across countries, as summarised in Table 3, any change in carers can be stressful, unsettling, and potentially traumatic, both in the short and long term, for the children, but also for their carers. Thus, for example, the evidence highlights the importance of the nature and effects of primary trauma for children, as well as secondary trauma for alternative carers, e.g., foster carers, who are exposed to the traumatic histories of the children [66]. The effects of this, however, are compounded when there is an intervention by the courts.

Boulay et al. [67] found that being placed on a judicial order—as is often required for children deprived of their parents—presents a potential risk factor for children’s development. Removal from parental family care by way of court recommendations may require a total break in the child’s relationship with their family care environment, compounding the negative biological and psychological effects of abuse and neglect, with concomitant detrimental impacts on the process of ‘becoming an adult’. This may cause life-long psychopathological disorders [67,68], with such vulnerabilities being exacerbated with the child being more intensely involved in the process of rupture [69]. The effects of disruption and changes for children who are part of legal proceedings and moved to unknown carers are known to make them more prone to problems with health, mental health and well-being, educational attainment, relationships, and potential trauma later in life [9]. In addition, Datta et al. [70] found that children who experience family breakdown can have a wide range of behavioural problems, with those living within conditions of ongoing trauma and stress possibly being most significantly affected. While such trauma id as relevant in domestic cases as in international ones, there are added complications and delays in confirming equivalent orders in destination countries and recognising the legal rights of the new carers.

## 6. Establishing the Significance of the Issue: Data

Underpinning the analysis of effective decision-making for the best interests of children, who have family in other countries, is access to data regarding the decision-making process and its outcome. Data and research concerning outcomes for children placed with family overseas are woefully inadequate in the UK, if not worldwide.

The UK has a requirement to report to the UN Committee on the Rights of the Child about progress and actions being undertaken to ensure rights are respected and realised for all children. The Children and Young People’s Commissioners of all four of the UK’s nations are involved in UNCRC reporting, and they rely on the robustness of reporting from responsible agencies, such as children’s services teams or the Department for Education. The Department for Education publishes annual statistics on looked-after children, including information about the reasons for children leaving care and system performance. The Adoption and Special Guardianship Leadership Board publishes quarterly data reports on system performance, including local authority-level data on the amount of time children wait to be placed with adoptive families, adopter recruitment, and the characteristics of children waiting for adoption. The Children and Family Court Advisory Service (Cafcass) also reports monthly on the number of care order applications made by councils. Cafcass has recently begun to track children placed with family overseas in public law cases. However, none of these institutions consistently track whether children have had family overseas considered in long-term permanency planning, nor do they publish data regarding children placed with family overseas.

The English Children’s Commissioner 2020 Periodic Review found that the ‘UK government does not prioritise children’s rights or voices in policy or legislative processes. There is a lack, inconsistent or incorrect use of, and/or poor quality of Child Rights Impact Assessments (CRIA) in all jurisdictions.’ [71]. It also found that ‘There is a lack of coherent, consistent, transparent, and systematic, disaggregated data collection concerning children across all jurisdictions, making it difficult to monitor and measure children’s needs and assess the fulfilment of their rights.’ [71]. None of these reports cover the extent to which a child’s best interests are upheld when culture, identity, roots, or family are located in other countries—other than for asylum-seeking or refugee children.

Although data collection and reporting is a challenge, CFAB has combined information obtained through Freedom of Information requests with its own study on international kinship care cases (Children and Families across Borders [72,73,74] to understand UK social work practice. In August 2021, CFAB sent a number of Freedom of Information (FOI) requests to 211 local UK authorities to understand the care provided for looked-after children with family members abroad, between January 2018 and December 2020 inclusive [75]. This request received an overall 94% response rate. The first FOI investigated the total number of looked-after children who had family members outside the UK. The second FOI concerned the number of looked-after children who were placed outside the UK. The final FOI explored the number and type of placement orders used when placing looked-after children overseas. The results were then compared with CFAB’s FOI request responses (2018), which covered the same areas of exploration during the period of January 2015–December 2017 inclusive [76].

The 2018 CFAB study included a case file audit of 200 CFAB cases referred between 2015 and 2016 [72]*,* with a purposive sample comprised of 100 cases involving exploration of potential kinship placement abroad for children [51], and 100 cases of child welfare and child protection issues [73]. The aim of this study was to understand the challenges and outcomes for children with family overseas in order to ensure their welfare. The CFAB study examined the effectiveness of international kinship care placements where public bodies such as courts and social workers are involved. Such moves can be very disruptive and challenging for both the children and the families involved in both countries, who may be distressed because of the changes in their care being made for them, which is a natural feature of such changes in any event. but adds extra layers of stress for them because of the transnational, transcultural, and trans-legal issues involved.

Two teleconference focus groups were subsequently conducted to help to test and explain the preliminary findings of the case audits. Research in Practice, an external organisation, facilitated the focus groups. Social workers and solicitors of local authority, as well as CAFCASS children’s guardians, were invited to participate. Professionals from four local authorities and CAFCASS were represented. Three professionals participated in the first focus group, and five in the second focus group. Four key themes were explored in the discussions:

Whether there were concerns for children in need of protection who travelled abroad from the UK;
(1)How viable the exploration of international family placements is for children in care;(2)The extent to which a child’s nationality is considered in case management;(3)The scope of consideration of the Brussels IIa Regulation or 1996 Hague Convention 2.


## 7. Discussion: Does Practice Reflect the Guidance?

Social work in the UK is based on respect for the inherent worth and dignity of all people, as expressed in the United Nations Universal Declaration of Human Rights, 1948. Over the course of their training, UK social workers learn how to build on identified strengths in families. However, a social worker’s ability to identify strengths, and, indeed, to maintain the best interest of the child, is fractured when viewed through different cultural and legal frameworks. The result, as illustrated in the sections below, is that few social workers even attempt to assess a child’s best interest in relation to a foreign family member or culture—let alone recommend that the child be placed with family overseas. Those who do find it is in a child’s best interest to join relatives in another country seem to have ongoing concerns about the long-term outcomes for the child.

In terms of the needs of children living in substitute families, a study of 96 adoptive parents in Wales found that the main support needs of families were promoting children’s health and development; fostering children’s identities; strengthening family relationships; providing financial and legal assistance; and managing contact with birth parents and significant others [77]. Such support is also required for children who join extended family overseas, with the added issues of cultural and legal differences for families and social workers to take into account, as addressed in this article. These areas of concern and evidence highlight the need for rigorous post-placement planning to be considered as part of the placement plan itself before the child joins new carers. Consideration should also be given to plans for managing contact with significant others in the UK that includes how such contact should take place (i.e., by phone call, direct contact, video call, or email). Again, good practice would require that any international placement plan should identify what authority (UK or overseas) is responsible for support, and would clarify support the expectations of the carer, e.g., school tuition fees or medical insurance.

Given the complex considerations of differing cultural norms and legislative contexts, child protection professions should liaise with counterparts—either statutory or private—in the destination country to request that visits be undertaken after the placement to identify any further support needed by the carers or the child.

## 8. Upholding Best Interests across Differing Cultural Norms and Values

The practice of upholding best interests across countries with differing cultural norms is a challenge. How do social workers balance their professional obligations, within the Children Act 1989 and the UNCRC, with their practical concerns about safeguarding a vulnerable child who is leaving their jurisdiction? The CFAB focus group findings [72] suggested that there are ongoing concerns for the protection of children who have travelled abroad from the UK (even where the child was judged as ‘safe’), as well as a lack of knowledge of the long-term outcomes for children placed abroad.

Participants identified concerns about the viability of placements in countries where there would be limited social services to support families or provisions for children in the event of placement breakdown. There was also some confusion as to the different approaches and effectiveness of transnational assessments. Some local authorities sent social workers abroad to conduct assessments, while others used ‘in country’ agencies, highlighting the possible presence of disparities between assessments of local social workers in the prospective carer’s country as compared to the knowledge, skills, and cultural competence of social workers in the receiving country. Sometimes, carers were invited to the UK for supplementary assessments, whilst some assessments used technology such as Skype to undertake these assessments.

CFAB’s 2022 findings [75] reflect the concerns voiced in the focus groups. Less than half (39%) of UK local authorities attempted to find family overseas for children in care. This was a decrease from 49% in 2018 [76]. For the period of 2018–2020, an estimated 233 children had family abroad explored as potential carers, a rise from the 202 found in the previous FOI request. Although fewer local authorities reported exploring family abroad, more children were supported in accessing their right to family than in previous years. Approximately 112 children were placed with family members abroad in 2018–2020, which is 9 fewer than in 2015–2017. A decrease was expected, as in 2020, the pandemic resulted in many border closures. Exact figures could not be reported due to the requirement to maintain confidentiality when numbers were low.

## 9. Ensuring Support for Children Leaving UK ‘Corporate Parents’ to Join Family Overseas

The needs of children in kinship care will often be met by their carers. Some children, however, need additional direct help, and given the issues raised in this article, the need is even more dire in situations where such kinship placements are part of transnational placements. Hunt [78] found that carers need support in managing changed family relationships, with just under twenty per cent in their study reporting unmet needs in establishing and managing contact.

The extent of support offered for children leaving the UK care system is not known. A written parliamentary question posed to the Secretary of State for Education by Helen Hayes, MP, in October 2022 received the following response: “*The department does not hold data on the number of looked after children who have been placed overseas specifically with family members either on a kinship foster placement or on a special guardianship order, therefore we do not hold data on the allowances these carers receive.*” [79].

One element of the CFAB study [73] was the review of cases where placement abroad was potentially in the best interest of the child(ren) in care. Of the 33 cases for which an outcome was known, 21 resulted in placements of 27 children outside the UK. Decisions on the remaining 45 cases were unknown due to CFAB’s involvement having ended following referral to local authorities. Just under 10 per cent (2 out of 21) of the cases that were audited, where children had been placed with family abroad, experienced placement breakdown. In both cases, the main reason for placement breakdown was because the carers were not able to cope with the psychological and behavioural challenges of the children. The case audit also found that some UK local authorities were not following through on ongoing responsibility if placement broke down abroad, or when challenges arose in the placement.

There is no statutory or legal guidance clarifying the length of time for which local authorities retain responsibility for a child after an international placement is made. Depending on the order used to place a child, it is sensible for the sending and receiving authorities to agree a ‘settling in’ period for the child and carer, as well as to identify any additional challenges or needs [80]. Good practice would include the involvement of a local authority permanency team, with the possibility of financial support, for a longer period of time than would be expected in domestic placements. This is because domestic kinship placements are likely to have been previously tested, something which is not possible with family overseas. In England, under Special Guardianship Orders, a degree of responsibility by the placing authority remains for three years. Special Guardians may also benefit from access to therapeutic funding through the Adoption and Special Guardianship Fund. Given the local authorities’ duty of care to the child, and given the additional risks associated with placing a child in another jurisdiction, a sending authority may need to be involved for a longer period of time. Such matters should be detailed in the placement plan.

## 10. Challenges in Applying National Legal Frameworks in International Family Settings

There were notable differences in the approaches of local authorities to assessing international placements, the legal orders used, and the post-placement support provided in the CFAB study [72,73,74]. Adherence to international regulations was found to have a positive impact on case management in relation to international placements, and it was important that social workers understood which domestic legal orders would be ‘relevant’ in another country or what other legal arrangements would need to be made. However, the CFAB findings suggest that, along with knowledge regarding comparative welfare, there is little professional education about migration and the specifics of legal provisions affecting children who cross international borders, including issues relating to the possible loss of a child’s identity and/or permanency in placements. Both the 1996 Hague Convention [81] and the Brussels lla Regulation of Europe [82] enable cross-country collaboration on children’s cases and set out provisions for determining the legal jurisdiction of cases where a child is present in one country but habitually resident in another. They also allow for legal orders made in one country to be recognised in another signatory country [82].

CFAB’s 2022 FOI request responses [75] reflected the practical challenges of applying English legal orders in international settings. Their third FOI request centred on what legal orders were used by children’s services teams in England and Wales to place looked-after children with family overseas in the period of 2018–2020. They found that 41% of authorities were unable to specify which orders were used in international placements. Special Guardianship Orders (SGO), a type of order which is a uniquely English creation, represented 33% of the reported orders. While this is a decrease from their 2018 FOI request, which saw SGOs representing 45% of all legal orders used, questions remain about the recognition of these orders across jurisdictions as SGOs do not exist in Scotland or Northern Ireland let alone in most other countries. This makes it difficult to mirror the rights and responsibilities intended for carers in receiving countries. Additionally, CFAB found that 16% of all orders used to place looked-after children were Interim Care Orders (ICOs). This raises questions about the ability of the sending local authority to exercise rights over the child once the child is outside of their jurisdiction.

A relevant example of the importance, and complicated nature, of considering of legal orders across jurisdictions involves children in care in the UK who are to be placed with family in Poland due to SGOs. According to the 1996 Hague Convention [58] and the Council Regulation Brussels Il Bis (EC) No 2201/2003 art. 56 [83], the procedures for consultation/obtaining the consent shall be governed by the national law of the responding State. However, local interpretation varies. The case of Poland illustrates the complicated nature of cross-border placements, even though they are members of both the 1996 Hague Convention and the Brussels II Bis. In Poland, the process of gaining consent for placement is completed by the competent court, depending on the locality of the carers. Once the order is made, the carer is able to exercise care over the child and secures access to the relevant benefits and support. If an SGO, for example, is made, this procedure ensures that the placement will be treated as a kinship foster placement. When the UK court seeks a kinship placement in Poland, a separate hearing should be arranged at least one month prior to the final hearing to allow time for the request to be heard in the relevant Polish court. The order requesting acceptance must be accompanied by a draft pro forma Special Guardianship Order, with a supplement outlining the meaning and the effect of the SGO in English domestic law. Additionally, the assessment of the potential carer, the threshold document or the fact-finding judgement, the birth certificate of the child/children, and the declaration from the prospective carers about their willingness to care for the child/children until the age of 18 are required.

Given the complicated nature of reunifying children with family overseas, it is fundamental that authorities understand how effective their procedures are and the ultimate benefit to the child for whom they have a duty to care. While statutory agencies carry the responsibility for publishing data regarding why children enter care, why they leave, or how long they wait to be placed, none of the statutory agencies in the UK openly publishes data on the frequency or effectiveness of international kinship placements. This is the gap that the CFAB studies and FOIs [72,73,74,75,76] are attempting to fill. The challenge of the data gap, is summarised in Table 4.

## 11. Establishing the Significance of the Issue: Data

When a child is taken into care, and the state assumes parental responsibility for them, a mapping of their extended family should be undertaken, and potential carers should be explored regardless of their geographic location. However, often this does not occur due to social worker concerns (see Table 5). It is apparent across the literature and in the CFAB study that there is a lack of data on children with family members overseas and children placed with family members overseas. This was confirmed by the CFAB 2022 FOI, which showed that over half of the local authorities (61%) either could not provide or reported not to have a response to the question of the number of children for who, overseas kinship carers had been explored [75].

It is striking that the cumulative number of children reportedly placed with family internationally from all four nations over a three-year period was 112. A similar request for data was made to Cafcass, the children and family court of England, and it was estimated that at least 130 children were placed by English local authorities with family in another country in the calendar year of 2021 alone. This raises the question of whether there has been a steep increase in England of children moved from the care of local authorities into international kinship placements, as well as the question of whether there are serious deficiencies in the authorities’ abilities to track and support international kinship placements.

The apparent lack of consistent data on children reunited with their families internationally is concerning given the very strong case for kinship care potentially being in the best interest of children combined with the multitude of risks inherent to international placements. It is unclear, and possibly unknown, how many children are affected by this. The aforementioned death of 8-year-old Victoria Climbie—a child in an international kinship placement—was both horrific and tragic, but it could have been prevented if there had been more effective interdepartmental sharing of data regarding children.

## 12. Conclusions

The CFAB report findings reported herein, as well as in the wider literature, indicates a range of key issues related to migration, different formulations of welfare, cultural differences, and the ways in which social work can determine the best practices in such situations. Social workers need to take all of these factors into account when assessing a child in need of protection and/or where placement with kin in another country might be an option. The welfare, or ‘the best interests of the child’, is a universal principle, but how this might be interpreted is likely to vary according to the customs and culture of each country, as well as their various economies, welfare systems, and legislation.

The CFAB study showed that overseas family placements are viable options. More than a third of cases in which an international family placement was explored resulted in such a placement. The actual placement rate may be higher, as the placement decision was unknown in 45% of cases. These findings add to the research findings outlined in this article concerning the value, as well as the challenges, of kinship care for children who are unable, for whatever reason, to reside with their parents. However, additional issues need to be taken into account; these include having to adjust to living in another country which the children may know only somewhat or not at all; the possible stresses for kinship carers, with possible other stresses due to additional factors of cross-country and cultural issues; and how contact with parents can take place while geographically distant from the kin carers/child(ren), see, e.g., [84].

The CFAB reports make recommendations for further developing a new practice, which has appeared in some local authorities, of inviting prospective family carers from abroad to come to the UK to be assessed by a UK social worker. This further raises issues of cultural competence from an understanding of cultural sensitivity, as discussed previously, and is used either as the main assessment or to supplement an assessment completed by a social worker overseas. This practice has been used in several cases and has been identified as something that ‘works well’ by the focus groups. MacAlister’s report [9] for the UK government advises that the many wider kin networks who care for their family members deserve, and need, ‘*far greater recognition, and support*’ [9]: 4. It also recommends that ‘*before decisions are made which place children into the care system, more must be done to bring wider family members and friends into decision making. This should start with a high-quality family group decision making process that invites families to come up with a family led plan to care for the child or children.*’ [9]: 4.

However, it is not usually possible with international placements to easily achieve such preplanning, assessments, and face-to-face family group decision-making processes, though it is possible that online platforms such as Zoom/MS Teams may aid in such international meetings going forward. Nevertheless, the possibility described herein of inviting prospective family carers from abroad to come to the UK to be assessed by a UK social worker may be informative.

Consequently to these issues, the CFAB reports recommend that local authorities and courts consider that a potential carer’s possible lack of familiarity with the child’s psychological and behavioural challenges can pose a risk to the long-term sustainability of any such placement, but, perhaps, particularly in international placements. In addition, consideration needs to be given to how well potential carers know the child, considering that they have been raised in a different country, as well as to their knowledge of the types of support needed by both carers and children over time to address the issues that are likely to arise.

Within these concerns and positives, there is also, in general, a lack of understanding of long-term outcomes for children placed abroad. Focus group participants expressed concerns that may not be informed by difficulties with the placement, or placement disruption, when a child is placed overseas. This is a significant risk, as children are sometimes placed abroad based on assessments completed by overseas social workers that have different requirements to UK assessments, and as children are placed in countries with limited social and family support services.

Social workers with good knowledge of transnational casework—whether employed by the state or as part of specialist agencies—therefore have an important role to play in developing comparative knowledge regarding conditions and services across national borders, informing professional education and participating in research, and contributing to formulating international policies and practice guides in particular fields. Specialist agencies and international associations also have their parts to play in lobbying national governments or international bodies under which conditions and practices in particular countries ignore or transgress human (including children’s) rights, and/or social workers are placed in uncomfortable or even risky positions with regard to practising forms of social work which legally and ethically adhere to children’s rights and best interests.

## Figures and Tables

**Table 1 children-10-00537-t001:** Foundational framework for kinship care.

The United Nations Convention on the Rights of the Child (UNCRC) has been ratified by 196 states, and is currently the most widely adopted international human rights law instrument. It requires those states to take measures by which *‘In all actions concerning children, whether undertaken by public or private social welfare institutions, courts of law, administrative authorities or legislative bodies, the best interests of the child shall be a primary consideration.’* [22]: article 3, including where children may need kinship care overseas. England’s Children Act 1989 requires children’s care agencies and their social workers to give preference to placements with kin carers. Research points consistently to a number of advantages of kinship care placements over alternatives such as foster care. Outcomes in terms of various child related matters, e.g., mental health and feelings of security, are better with kinship care in general. There is a much greater likelihood that the children will know, to some extent, their proposed new carers, and/or the new carers will know the parents/family of origin.

**Table 2 children-10-00537-t002:** The Welfare Checklist as per Section 1 of the Children Act 1989.

*Welfare Checklist as per Section 1 of the Children Act 1989, an essential tool for social workers assessing the best interest of the child.* The ascertainable wishes and feelings of the child concerned (considered in the light of age and understanding); physical, emotional and educational needs; the likely effect on the hcild of any change in circumstances; age, sex, background and any characteristics of the child which the court considers relevant; any harm which the child has suffered or is at risk of suffering how capable each of the parents, and any other person in relation to whom the court considers the question to be relevant, is of meeting the child’s needs; the range of powers available to the court under this Act in the proceedings in question.

**Table 3 children-10-00537-t003:** Judicial and policy challenges across different jurisdictions.

A key challenge in enacting the United Nations Convention on the Rights of the Child, which states that ‘In all actions concerning children the best interests of the child shall be a primary consideration’, is in how this is assessed and judged in terms of children in need of substitute care, and in situations that may involve a transnational kinship placement. There can be complications and delays in confirming equivalent orders in destination countries and recognising the legal rights of the new carers. The UNCRC still allows many variations in national policy and law in terms of how best interests can be determined, and the effects of this on the welfare and protection of children and young people.

**Table 4 children-10-00537-t004:** Importance of information and data.

None of the statutory agencies in the UK—e.g., courts and children’s services agencies—openly publish data on the frequency or effectiveness of international kinship placements, and these agencies need to be more robust and open in publishing these data. Such data collation and publication is important in order to monitor, review, and develop the work of courts and agencies concerning their duties to ensure the best interests of the child, as required under the UNCRC guidelines on possible and actual international kinship placements. These data should include the frequency and effectiveness of services for children, in addition to: - Why the children enter care/may need such placements; - How long they wait to be placed; - Why they leave; - What follow-up is planned to take place in the receiving country; - What the outcome is for these children, as compared to other types of placements

**Table 5 children-10-00537-t005:** Practice implications.

The CFAB project found that professionals were concerned about: - A lack of understanding of long-term outcomes and follow up for children placed abroad; - A lack of information on difficulties within the placement when a child is placed overseas; - Possible risks for children who may be placed abroad based on assessments by overseas social workers that have different requirements to UK assessments; - The fact that children can be placed in countries with limited social and family support services.

## Data Availability

Not applicable.

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
