# Peer review of "Applying Universal Principles of ‘Best Interest’: Practice Challenges across Transnational Jurisdictions, Cultural Norms, and Values"

_children, 2023, doi:10.3390/children10030537_

Round 1

Reviewer 1 Report

The article takes up a very complex and marginally discussed topic where research is needed. The authors ask how far universal principles can be applied in culturally diverse contexts and what exactly this means for social work practice. The contribution vividly describes how cultural norms influence the actions of social workers, a circumstance that can be particularly challenging in international contexts. A similar picture emerges from the different legal systems in the different countries, some of which are taken up as examples. Overall it is a very interesting and relevant contribution for social work practice and theory. For the journal children your article connects to the aim of interdisciplinary research as it does not deal with pediatric topics in a narrow sense, but refers to transnational social work with children. 

It is important to revise this paper systematically before publication. Suggestions summarized in keywords:

- the choice of headings and sub-headings is not always comprehensible. e.g. Chapter 2 "Materials and Methods" does not explain methods, only chapter 3 "Results" starts to talk about methodology.  It would improve the quality of the paper if you could provide more coherence between headings and content throughout the document. 

- methodology: Please clarify the connections between case study and results. e.g. It is not clear which findings refer to the case study with 200 case file audit of CFAB (2015-2016) and which refer to the 79 case file audit of CFAB (please indicate the year?) or why only the results from the 21 cases discussed in the focus group are relevant for the article? Is the CFAB study with 78 cases a different one? Please clarify. 

- Line 261 please find a clearer formulation  “would appear to be….

- Line 261 to 278 frequent repetition of the word “should” – are these recommendations? Recommendations should rather be formulated in the conclusive part

- From line 443: to me it is no clear whether these are results or recommendations 

- Line 461 and Line 499 repeats the same sentence

- Line 131 "Roman et al" reference missing in the literature

- Line 134 please check the formulation “family care kinship family care”

- Line 146 “Determining Best Interests Across Different Cultural Norms and Values” is this a heading?

- please proof-check the literature e.g. 26. Durrant - the year is missing

Author Response

Thank you very much for your time and expertise in reviewing this submission.

Please see our responses below/

‘It is important to revise this paper systematically before publication. Suggestions summarized in keywords:’- ‘the choice of headings and sub-headings is not always comprehensible. e.g. Chapter 2 "Materials and Methods" does not explain methods, only chapter 3 "Results" starts to talk about methodology.  It would improve the quality of the paper if you could provide more coherence between headings and content throughout the document.’ RESPONSE: Thank you for these points- these sections and others have been completely reworked from these comments, and this heading removed, and  with results now becoming ‘Discussion’.

‘methodology: Please clarify the connections between case study and results. e.g. It is not clear which findings refer to the case study with 200 case file audit of CFAB (2015-2016) and which refer to the 79 case file audit of CFAB (please indicate the year?) or why only the results from the 21 cases discussed in the focus group are relevant for the article? Is the CFAB study with 78 cases a different one? Please clarify.’  RESPONSE: Thank you for these points- this section along with others has been completely reworked and amended for clarity  from these comments. Please see  sections headed  ‘Establishing the significance of the issue Data;  Upholding Best Interests Across Differing Cultural Norms and Values;  Ensuring support for children leaving UK ‘corporate parents’ to join family overseas; Ensuring support for children leaving UK ‘corporate parents’ to join family overseas; and Challenges in applying national legal frameworks in international family settings’.

‘Line 261 please find a clearer formulation  “would appear to be….”’  RESPONSE:  this has been changed to  ‘These areas of concern and evidence highlight the need  for rigorous  post-placement planning to be undertaken and considered as part of the placement plan itself and conducted prior to the child being placed’,   now in new  line 350.

Re comment – ‘From line 443: to me it is no clear whether these are results or recommendations ‘  and the  related comment on use of the term ‘should’. RESPONSE: – We have made substantial   changes  across several sections to  make it clear what is being referred to is  current good practice, or are  findings- including e.g. in new  lines 350- 356. Re use of the term ‘should’, this has been amended, so e.g. in new lines 421-423,  ‘Good practice  would ensure these matters are detailed in the placement plan,  to include  an agreement as to the duration of the support including a mutually determined cut-off date which is suitable based on the child’s needs’.  Thank you.

Again related to these points from the reviewer and our responses on these,  re ‘Recommendations should rather be formulated in the conclusive part, and many sentences are not cited there is a lot of research available when keywords are entered therefore such sentence must be referenced for example informations in line 278-280’, RESPONSE:   we have moved a large section that talks about good practice and recommendations in terms of what  arises from our inquiries- so now separate from findings - into the Discussion section.  Thus we have changed around where certain sections to make clear where they are recommendations from the review of the literature and the work of CFAB  from their research, so this has now been addressed, and recommendations  are now in the  ‘Discussion’ section,  starting  on line 335,   thank you.

Line 461 and Line 499 repeats the same sentence’- RESPONSE:   This has now been  reworked, thank you

‘Line 131 "Roman et al" reference missing in the literature’.  RESPONSE:  This has now been added

‘Line 134 please check the formulation “family care kinship family care”.’  RESPONSE:  : this has been changed, thank you

‘ Line 146 'Determining Best Interests Across Different Cultural Norms and Values” is this a heading?’ RESPONSE:  Thanks for this- yes it is, and highlighted as such  in new line  127.

- ‘please proof-check the literature e.g. 26. Durrant - the year is missing’;   RESPONSE:  general proof reading carried out. and in relation to Durrant the year is now included- due to reworking, this is now reference 28, not 26; changed because of additions and reworking of the article so that the references are now in a different order.

Reviewer 2 Report

The paper deals with a well-known and relevant topic, and in the development of the argumentation the authors highlight the complexity of the cases and the different components that emerge as characteristics of this problems. The paper however, has an advocacy perspective rather than being  a scientific analysis. This is  evident in some aspects: the literature review, which could be further deepened, but, above all , the empirical analysis of the cases (paragraph 3). 

The empirical materials on which the analyses are conducted are not clearly presented; there is mention of a data collection done by CFAB between 2015 and 2016, but in the discussion of the results there is apparently no reference to these cases that would have been discussed during focus groups with solicitors and social workers. When did the focus groups take place? how many people participated? which parts emerged? The methodology of analysis is not mentioned. 

in the paragraph Challenges in applying national legal frameworks in international family settings, there is a repetition of the same sentence at the line 461-463 and the line  499-501.

Some other mistakes /typos or not finished sentences are ath the following lines:

-131

-134

-184 (please don't use contracted words)

-261

Since the conclusions and argumentations are convincing, it is suggested a general revision of the empirical result, better explaining what comes from the discussions in the focus groups and the materials collected. Instead, it appears more an advocacy-policy oriented work, without being such type of paper.

A minor proofreading is also requested. 

Author Response

Thank you very much for your time and expertise in reviewing this submission.

Please see our responses below.

‘The paper deals with a well-known and relevant topic, and in the development of the argumentation the authors highlight the complexity of the cases and the different components that emerge as characteristics of this problems. The paper however, has an advocacy perspective rather than being  a scientific analysis. This is  evident in some aspects: the literature review, which could be further deepened, but, above all , the empirical analysis of the cases (paragraph 3)’.  RESPONSE:  Thank you for these points- this section has been completely reworked from these comments, and re how it seems an advocacy piece, please see how we have moved  the  paragraphs/sentences from previous location to   the Discussion section,  where these  talk about good practice and what arises from our inquiries separate from findings .  Thus we have changed around where certain sections to make clear where they are recommendations from the review of the literature and the work of CFAB  from their research, so this has now been addressed, and recommendations  are now in the  ‘Discussion’ section,  starting  on line 335, thank you

‘The empirical materials on which the analyses are conducted are not clearly presented; there is mention of a data collection done by CFAB between 2015 and 2016, but in the discussion of the results there is apparently no reference to these cases that would have been discussed during focus groups with solicitors and social workers. When did the focus groups take place? how many people participated? which parts emerged? The methodology of analysis is not mentioned’.  RESPONSE:  Again, thank you for these points- this section has been completely reworked from these comments- please see sections headed  ‘Establishing the significance of the issue: Data, and Upholding Best Interests Across Differing Cultural Norms and Values,  Ensuring support for children leaving UK ‘corporate parents’ to join family overseas, Ensuring support for children leaving UK ‘corporate parents’ to join family overseas, and Challenges in applying national legal frameworks in international family settings’.

‘in the paragraph Challenges in applying national legal frameworks in international family settings, there is a repetition of the same sentence at the line 461-463 and the line  499-501’.  RESPONSE:  now removed, thank you

‘Some other mistakes /typos or not finished sentences are ath the following lines:’

-131 RESPONSE:  thank you, now amended

-134 RESPONSE:  thank you, now amended

-184 (please don't use contracted words)  RESPONSE:  thank you, now amended

-261’ RESPONSE:  thank you, now amended

‘Since the conclusions and argumentations are convincing, it is suggested a general revision of the empirical result, better explaining what comes from the discussions in the focus groups and the materials collected. Instead, it appears more an advocacy-policy oriented work, without being such type of paper’. RESPONSE:  Thank you for these points- this section along with others has been completely reworked and amended for clarity  from these comments. Please see  sections headed  ‘Establishing the significance of the issue: Data, and Upholding Best Interests Across Differing Cultural Norms and Values,  Ensuring support for children leaving UK ‘corporate parents’ to join family overseas, Ensuring support for children leaving UK ‘corporate parents’ to join family overseas, and Challenges in applying national legal frameworks in international family settings’.

‘A minor proofreading is also requested’. RESPONSE:  Thank you – this has been done

Reviewer 3 Report

In a way, the paper attempted to study the applicability of children ‘best interest’ principle across differing cultural norms and values. However, the manuscript articulation of the principle in the Title, Abstract, and Introduction seems not clear enough or well connected. In particular, I could not reconcile whether the Method of the study is a literature review and/or a legal document analysis?

Therefore, would the authors adjust the articulation of the Title, to reflect more clearly in the Abstract, and the Introduction? Would the authors adjust the presentation of the Methods to make room for a possible replicability? In other words, what literature was reviewed and how? What documents were reviewed and how? And the how of other methods included.

Author Response

Thank you very much for your time and expertise in reviewing this submission.

Please see our responses below.

‘In a way, the paper attempted to study the applicability of children ‘best interest’ principle across differing cultural norms and values. However, the manuscript articulation of the principle in the Title, Abstract, and Introduction seems not clear enough or well connected. In particular, I could not reconcile whether the Method of the study is a literature review and/or a legal document analysis?’… ‘Therefore, would the authors adjust the articulation of the Title, to reflect more clearly in the Abstract, and the Introduction? Would the authors adjust the presentation of the Methods to make room for a possible replicability?’… ‘In other words, what literature was reviewed and how? What documents were reviewed and how? And the how of other methods included. RESPONSE:  Thank you – this has been done. Also, please see  new lines 6-69, with details on these then referenced throughout the article. In terms of the Title, we thank you for this very valuable point , and have amended the Title to ‘Applying universal principles of ‘best interest’: practice challenges across transnational jurisdictions, cultural norms and values’.

Reviewer 4 Report

Dear Authors,

 The article has the qualifications to guide the field, other practitioners, researchers and policy makers. This is an original topic; however, the way of presentation is excessively disorganized. A major concern is the structure of the manuscript, including a particularly weak introduction section describing. Overall, there are general errors in writing style and references, these should be corrected.

Please find my comments and feedback below with reference to lines.

Sincerely,

Abstract

Line 20-22 ‘’Agency case records were analysed to identify a range of factors in relation to children placed with ‘kin’ carers across national borders relating to the understanding and application of legislation and policy in transnational contexts, and findings discussed within professional stakeholder focus groups’’- You should modified the word “analysed to” as “examined to’’. Because you are just the commentators.

Introduction

It would be good to see information about the importance of the topic in the first paragraph of the introduction. For this reason, the information in lines 41-52 should be in place of line 32-40 information.

Materials and Methods

line 56- There is no need for such a title as this is a review article. Essentially, the review article should have an introduction, sub-headings, conclusions and recommendations.

Line 76- I think it will be child(ren) instead of children(ren).

One-sentence paragraphs cause serious disconnections. For example; sentences of line 141-144 many sentences like this...

Line 146- Isn't it a subtitle?

footnote after line 197 should have been cited in sources.

The constant repetition of the purpose of the article is not understood (e.g line 146, line 376) Whereas there should be a clearly stated purpose at the end of the Introduction.

Many sentences are not cited.  There is a lot of research available when keywords are entered. Therefore, such sentences must be referenced. For example; informations in line 278-280.

Line 366-3.Results- There is no need for such a title.

line 376- ‘’The aim of the study was to better understand the outcomes for these children and 376 any challenges for ensuring their welfare’’. Doesn't the work mentioned here belong to CFAB? If so, there is no need for a new paragraph. sentence must continue from line 375.

Many short interconnected paragraphs throughout the entire text should be combined for topic integrity.

Line 393- Upholding Best Interests Across Differing Cultural Norms and Values- I don't think such a title is necessary. this section contains the findings of the study in the previous section. That is, the continuation of the other section.

Conclusions

Its valuable for the authors to make some comments on what this could mean for overseas family placements in UK. Hence, the conclusion section is enough.

Author Response

Thank you very much for your time and expertise in reviewing this submission.

Please see our responses below.

‘The article has the qualifications to guide the field, other practitioners, researchers and policy makers. This is an original topic; however, the way of presentation is excessively disorganized. A major concern is the structure of the manuscript, including a particularly weak introduction section describing. Overall, there are general errors in writing style and references, these should be corrected.

Please find my comments and feedback below with reference to lines.’ We have substantially reworked the piece to address these points, thank you.

Abstract

‘Line 20-22 ‘’Agency case records were analysed to identify a range of factors in relation to children placed with ‘kin’ carers across national borders relating to the understanding and application of legislation and policy in transnational contexts, and findings discussed within professional stakeholder focus groups’’- You should modified the word “analysed to” as “examined to’’. Because you are just the commentators’ RESPONSE:  Thank you – this has been done. please see  new lines  54-58, and 76-70.

Introduction

‘It would be good to see information about the importance of the topic in the first paragraph of the introduction. For this reason, the information in lines 41-52 should be in place of line 32-40 information’. RESPONSE:  Thank you – this has been done/

Materials and Methods

‘line 56- There is no need for such a title as this is a review article’. RESPONSE:   This has been amended, thank you

‘Line 76- I think it will be child(ren) instead of children(ren)’. RESPONSE:  thank you, now amended.

One-sentence paragraphs cause serious disconnections. For example; sentences of line 141-144 many sentences like this.’  RESPONSE:  thank you,  these have now amended as appropriate

Line 146- Isn't it a subtitle?’ RESPONSE:  thank you, no, we have now amended, so hope this is  now clear

‘footnote after line 197 should have been cited in sources’. RESPONSE:  thank you, now amended, and it is in the references

‘The constant repetition of the purpose of the article is not understood (e.g line 146, line 376) Whereas there should be a clearly stated purpose at the end of the Introduction’. RESPONSE:  thank you, now amended, and this has been made clearer is in the Abstract and Introduction

Many sentences are not cited.  There is a lot of research available when keywords are entered. Therefore, such sentences must be referenced. For example; informations in line 278-280’.  RESPONSE:  Thank you- looking again at this, we can see why this is confusing; these statements were actually in relation to good practice recommendations from the review carried out, and so these sections have been moved into the Discussion section rather than earlier in the article.

Line 366-3.Results- There is no need for such a title’. RESPONSE:  thank you, now amended

line 376- ‘’The aim of the study was to better understand the outcomes for these children and 376 any challenges for ensuring their welfare’’. Doesn't the work mentioned here belong to CFAB? If so, there is no need for a new paragraph. sentence must continue from line 375’. RESPONSE:  Amended, thank you.

Many short interconnected paragraphs throughout the entire text should be combined for topic integrity’. RESPONSE:   thank you, now amended as appropriate

‘Line 393- Upholding Best Interests Across Differing Cultural Norms and Values- I don't think such a title is necessary. this section contains the findings of the study in the previous section. That is, the continuation of the other section’. RESPONSE:   Thank you; We can see the point being made by the reviewer, but as this was the key element we were exploring in this article, we would still ask that this remains in there to highlight this again at this point in the article

Round 2

Reviewer 4 Report

Dear authors,

Thank you for all the revisions and clarifications.